# Mechanisms of hyperexcitability in Alzheimer's disease hiPSC-derived neurons and cerebral organoids vs isogenic controls

Swagata Ghatak[1†], Nima Dolatabadi[1†], Dorit Trudler[1], XiaoTong Zhang[1], Yin Wu[1], Madhav Mohata[1], Rajesh Ambasudhan[2], Maria Talantova[1], Stuart A Lipton[1,2,3,4,5]*

[1]Department of Molecular Medicine, The Scripps Research Institute, La Jolla, United States; [2]Neurodegenerative Disease Center, Scintillon Institute, San Diego, United States; [3]Department of Neuroscience, The Scripps Research Institute, La Jolla, United States; [4]Neuroscience Translational Center, The Scripps Research Institute, La Jolla, United States; [5]Department of Neurosciences, School of Medicine, University of California, San Diego, San Diego, United States

**Abstract** Human Alzheimer's disease (AD) brains and transgenic AD mouse models manifest hyperexcitability. This aberrant electrical activity is caused by synaptic dysfunction that represents the major pathophysiological correlate of cognitive decline. However, the underlying mechanism for this excessive excitability remains incompletely understood. To investigate the basis for the hyperactivity, we performed electrophysiological and immunofluorescence studies on hiPSC-derived cerebrocortical neuronal cultures and cerebral organoids bearing AD-related mutations in presenilin-1 or amyloid precursor protein vs. isogenic gene corrected controls. In the AD hiPSC-derived neurons/organoids, we found increased excitatory bursting activity, which could be explained in part by a decrease in neurite length. AD hiPSC-derived neurons also displayed increased sodium current density and increased excitatory and decreased inhibitory synaptic activity. Our findings establish hiPSC-derived AD neuronal cultures and organoids as a relevant model of early AD pathophysiology and provide mechanistic insight into the observed hyperexcitability.

*For correspondence:
slipton@scripps.edu

†These authors contributed equally to this work

Competing interests: The authors declare that no competing interests exist.

## Introduction

Emerging evidence suggests that patients with Alzheimer's disease (AD) manifest an increased incidence of neuronal hyperactivity, leading to non-convulsive epileptic discharges (*Lam et al., 2017*; *Vossel et al., 2013*). These patients also display a faster rate of cognitive decline consistent with the notion that the aberrant activity is associated with disease progression. Moreover, both sporadic (S) and familial (F) AD patients show neuronal hyperactivity, with onset during the initial stages of the disease (*Palop and Mucke, 2009*; *Palop and Mucke, 2016*). Mutations in amyloid precursor protein (APP) or presenilin (PSEN or PS) genes 1/2, which increase amyloid-β (Aβ) peptide, cause dominantly inherited forms of the disease (*Woodruff et al., 2013*). These patients show increased activation in the right anterior hippocampus by functional MRI early in the disease (*Quiroz et al., 2010*). Moreover, both humans with AD and AD transgenic models, including hAPP-J20 and APP/PS1 mice, manifest non-convulsive seizure activity/spike-wave discharges on electroencephalograms (*Nygaard et al., 2015*; *Verret et al., 2012*; *Vossel et al., 2013*).

While AD transgenic animal models have been used extensively to study the mechanisms of the disease (*Palop and Mucke, 2016*; *Šišková et al., 2014*) the electrophysiological basis of the

observed hyperexcitability still remains incompletely understood. The recent advent of human induced pluripotent stem cell (hiPSC)-derived neurons affords the unique opportunity for monitoring pathological electrical activity and underlying mechanisms in a 'human context,' and on a patient-specific genetic background. For example, recent studies have shown increased calcium transients in a cerebral organotypic hiPSC-derived culture system bearing FAD mutations (*Park et al., 2018*). However, there remains a lack of electrophysiological characterization of disease phenotypes in neurons derived from hiPSCs carrying FAD mutations. It should be acknowledged that abnormal circuits related to aberrant electrical activity in AD brains might not be completely replicated in reductionist hiPSC-based preparations even though our 2D cultures contain both excitatory cerebrocortical neurons and inhibitory interneurons, and our 3D cerebral organoids show clear cortical layer formation. Importantly, however, abnormal neuronal morphology, disrupted ion channel properties, and synaptic dysfunction underlying aberrant electrical activity are all retained in these hiPSC-derived preparations compared to more intact systems, and are therefore studied in some detail here. In fact, evidence from both human AD brain and transgenic AD mouse models suggests that changes in channel properties and neurite length similar to that observed here may indeed be involved in the altered electrical excitability (*Kim et al., 2007*; *Palop and Mucke, 2016*; *Šišková et al., 2014*).

In the present study, we examine the electrophysiological properties of cerebrocortical cultures derived from three separate AD-like hiPSC lines bearing PS1 or hAPP mutations (vs. their gene-corrected isogenic wild-type (WT) controls): (i) PS1 ΔE9, a point mutation in the splice acceptor consensus sequence of intron eight that impairs γ-secretase-dependent functions of PS1; (ii) PS1$^{M146V}$, a mutation in transmembrane domain 2 of PS1 that occurs frequently in FAD patients; (iii) APP$^{swe}$, double missense mutation in the APP gene leading to increased Aβ production (*Paquet et al., 2016*; *Woodruff et al., 2013*). We find abnormal electrical activity in both AD hiPSC-derived 2D cultures and 3D cerebral organoids compared to their respective isogenic controls. Moreover, our work provides mechanistic insight into atypical neuronal morphology, altered ion channel physiology, and disrupted synaptic function underlying the aberrant electrical activity observed in these AD hiPSC-derived cortical neurons.

## Results

### AD hiPSC-derived neurons show enhanced excitability compared to isogenic WT neurons

Initially, we studied 2D cultures of AD hiPSC-derived cerebrocortical neurons ('AD neurons') with one allele bearing the PS1 ΔE9 mutation (ΔE9/WT), PS1$^{M146V}$ mutation (M146V/WT), or APP$^{swe}$ mutation (APP$^{swe}$/WT) vs. isogenic WT/WT controls (*Figure 1—figure supplement 1A*). Starting at 5 weeks in culture, AD neurons displayed the forebrain marker, FOXG1, and cortical neuronal marker, CTIP2 (*Figure 1—figure supplement 1B,C*) (*Woodruff et al., 2013*). The PS1 mutants (ΔE9/WT, M146V/WT) manifested an increase in the Aβ$_{42}$:Aβ$_{40}$ ratio, while the APP$^{swe}$ mutant showed an increase in the level of total Aβ (*Figure 1—figure supplement 1D,E*). Both PS1 and APP$^{swe}$ mutant AD neurons showed a 2–3-fold increase in the frequency of spontaneous action potentials compared to cerebrocortical neurons derived from their respective gene-corrected isogenic controls (WT/WT; *Figure 1A,B*, *Figure 1—figure supplement 2A,B*). Additionally, we found a marked increase in evoked activity in AD neurons vs. their isogenic controls. AD neurons fired multiple action potentials in response to current steps in contrast to a single or at most a few spikes elicited in WT neurons (*Figure 1C* and *Figure 1—figure supplement 2C*). Although we did not find differences between the resting membrane potential and action potential firing threshold between AD and control neurons (*Figure 1D,E*, *Figure 1—figure supplement 2D,E*), action potential height was significantly greater and half-width was significantly smaller in AD hiPSC-derived neurons compared to their respective WT controls (*Figure 1F,G*, *Figure 1—figure supplement 2F,G*). AD neurons with narrow spikes also fired at higher frequencies with faster decay times (*Figure 1H*, *Figure 1—figure supplement 2H*).

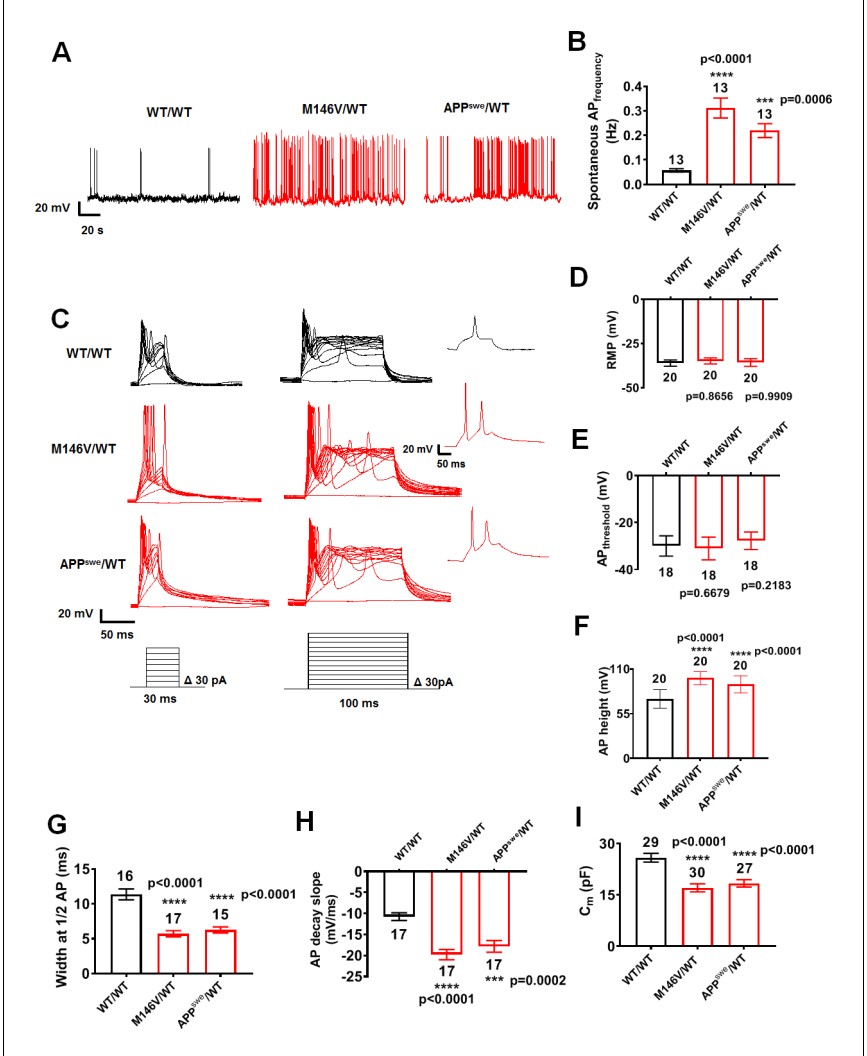

**Figure 1.** AD neurons show enhanced excitability compared to isogenic control neurons. (**A**) Spontaneous action potentials (sAP) at resting membrane potential (RMP). WT/WT hiPSC-derived cerebrocortical neuron data in black, M146V/WT and APPswe/WT in red. (**B**) Quantification of sAP frequency. (**C**) Evoked APs in neurons hyperpolarized to −60 mV. Single traces (insets). (**D–H**) Neuronal membrane and AP properties. Quantification of resting membrane potential (RMP, (**D**), AP threshold (AP$_{threshold}$, (**E**); AP height (**F**); width at AP half height (**G**); AP decay slope (**H**). (**I**). Quantification of cell capacitance (C$_m$), reflecting neuronal size. Data are mean ± SEM. Statistical significance analyzed by ANOVA with post hoc Dunnett's test. Exact p values for comparison to WT are listed in the bar graphs in this and subsequent figures. Unless otherwise stated, total number of neurons quantified is listed above the bars in this and subsequent electrophysiology figures.

The online version of this article includes the following source data and figure supplement(s) for figure 1:

**Source data 1.** Excel files containing data shown as summary bar graph in *Figure 1B,D–I*.

**Figure supplement 1.** AD hiPSC-derived neuronal cultures express cortical neuronal markers and aberrant Aβ levels compared to isogenic controls.

**Figure supplement 2.** ΔE9/WT neurons show enhanced excitability compared to isogenic control neurons.

## AD hiPSC-derived neurons manifest shorter neuritic processes and altered sodium channel properties compared to isogenic WT neurons

To begin to explain the underlying mechanism for the increased spontaneous and evoked firing properties of AD neurons, we found that these neurons manifest smaller cell capacitance, indicating a more compact overall electrical 'size' compared to WT (*Figure 1I*, *Figure 1—figure supplement 2I*). Consistent with this observation, we also found a decrease in total area covered by neurites in

the AD neurons, as measured histologically by anti-β-tubulin III (Tuj1) and anti-microtubule associated protein 2 (MAP2) antibody labeling, while somal size was not affected (*Figure 2A–C*, *Figure 2—figure supplement 1A,B*, *Figure 2—figure supplement 2A–C*). To further investigate this altered neuronal morphology, we transfected the neurons with green fluorescent protein (GFP). Only 10–20% of the cells expressed GFP, thus allowing us to trace the transfected neurons. We found shorter neurites and decreased branching in AD neurons compared to isogenic WT controls (*Figure 2D–F*).

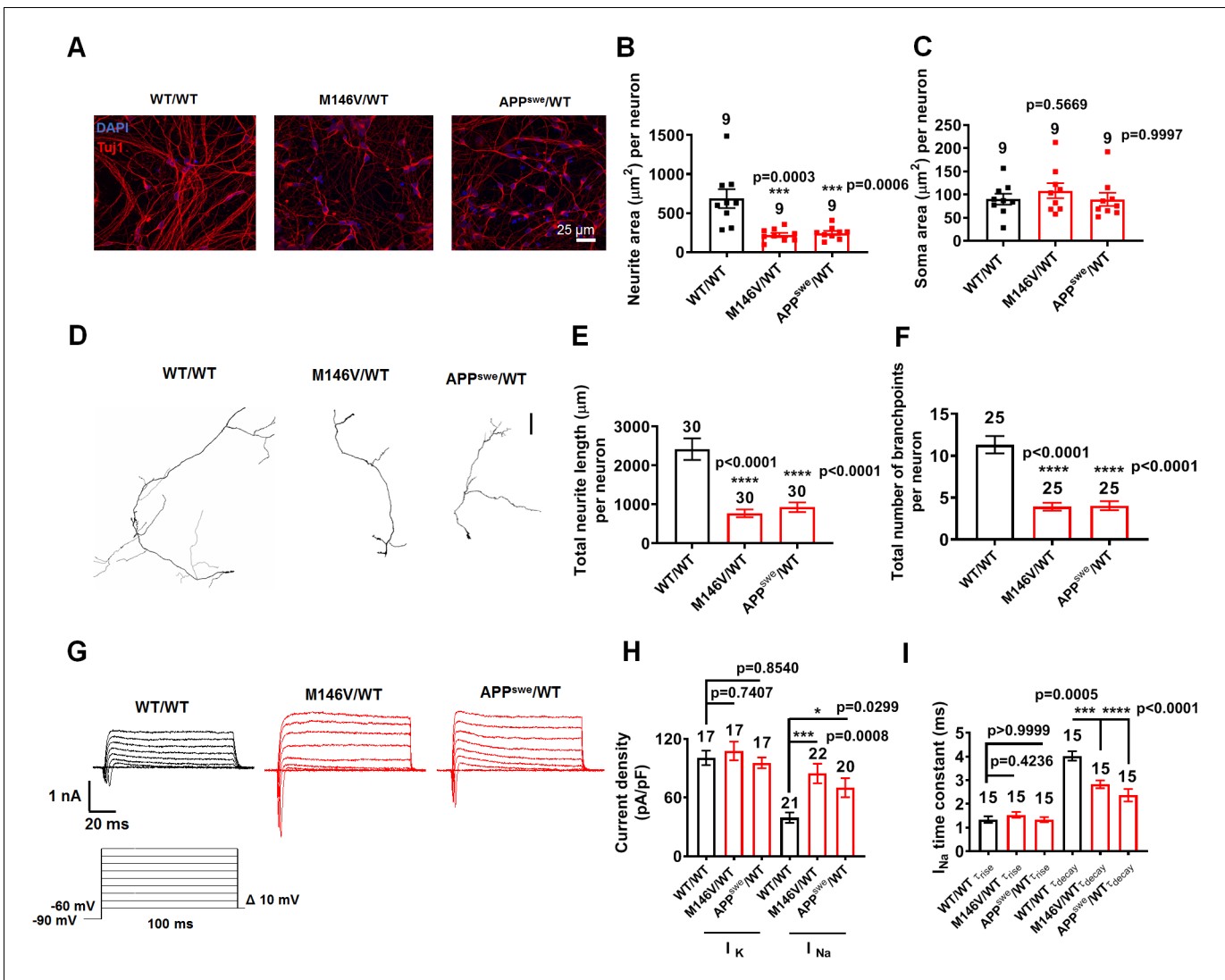

**Figure 2.** AD neurons show differences in morphology and sodium channel properties compared to WT neurons. WT/WT hiPSC-derived cerebrocortical neuron data in black, M146V/WT and APP^swe/WT in red at 5 weeks of culture. (**A**) Representative images of cells expressing the neuronal marker Tuj1. (**B**) Quantification of area covered by neurites expressing Tuj1 normalized to total number of neurons. (**C**) Quantification of somal area normalized to total number of neurons. Total number of random fields of neuronal cultures analyzed in three separate experiments is listed above the bars in B,C. (**D**) Representative tracings of neurites from WT/WT, APP^swe/WT, and M146V/WT AD lines. Scale bar: 100 µm. (**E**) Quantification of total neurite length. (**F**) Quantification of total number of neurite branchpoints. Total number of neurons analyzed in three separate neuronal cultures is listed above the bars in E,F. (**G**) Representative sodium and potassium currents recorded from neurons clamped at −70 mV. (**H**) Current densities. (**I**) Sodium current ($I_{Na}$) rise time ($\tau_{rise}$) and decay time ($\tau_{decay}$). Data are mean ± SEM. Statistical significance analyzed by ANOVA with post hoc Dunnett's test. The online version of this article includes the following source data and figure supplement(s) for figure 2:

**Source data 1.** Excel files containing data shown as summary bar graph in *Figure 2E,F,H,I*.
**Figure supplement 1.** Changes in MAP2 and Tuj1 expression.
**Figure supplement 2.** Changes in Tuj1 expression and sodium channel properties in ΔE9/WT neurons.

To elucidate possible differences in current density that could underlie the increased excitability of AD neurons, we next monitored their sodium and potassium currents (*Carter and Bean, 2011*). We found significantly greater sodium current density in AD neurons compared to WT neurons after 5 weeks in culture, while potassium current density remained unchanged (*Figure 2G,H*, *Figure 2—figure supplement 2D,E*). Additionally, we found a significantly faster decay constant for the sodium current in AD neurons compared to isogenic WT controls, while the rise times did not differ (*Figure 2I*, *Figure 2—figure supplement 2F*). This finding supports the notion that a more rapid recovery of sodium channels from inactivation in AD neurons would increase channel availability after the spike, thus reducing the refractory period and facilitating more rapid repetitive firing (*Carter and Bean, 2011*). Potentially underlying these effects, previous studies found that $A\beta_{42}$ oligomers or APP result in an increase in surface expression of sodium channel Nav1.6 in AD models, both in vitro and in vivo (*Ciccone et al., 2019*; *Liu et al., 2015*; *Wang et al., 2016*). Additionally, β-secretase 1 (BACE1) and PS1/γ-secretase-mediated processing of Navβ2 has been reported to enhance surface expression of this sodium channel, as observed in AD animal models and in human patients (*Hu et al., 2017*; *Kim et al., 2005*; *Wong et al., 2005*). Notably, β-subunits have previously been shown to increase current amplitudes and accelerate current decay kinetics (*Aman et al., 2009*; *Lopez-Santiago et al., 2006*; *Zimmer and Benndorf, 2007*).

## PS1 mutant AD neurons show developmental differences at early timepoints compared to WT neurons

To rule out the possibility that a lag in development in the WT hiPSC cultures compared to the AD cultures might account for the above findings, we studied neuronal electrical properties over time. Indeed, we found that at 2 weeks in culture, the PS1 mutant AD neurons (M146V/WT and ΔE9/WT) manifested greater sodium and potassium current densities, increased electrical capacitance, and increased synaptic density measured histologically compared to their gene-corrected isogenic WT counterparts (*Figure 3A–E*, *Figure 3—figure supplement 1A–D*, *Figure 3—figure supplement 2A, B*). These results possibly reflect a faster maturation pattern of PS1 mutant AD neurons than WT after 2 weeks in culture. By 4 weeks, however, AD neurons displayed similar potassium current densities and synaptic density compared to their isogenic WT controls; however, AD neurons now manifested a decrease in electrical capacitance (*Figure 3A–E*, *Figure 3—figure supplement 1A–D*, *Figure 3—figure supplement 2A,B*). The decrease in electrical capacitance is consistent with significant loss of neurites in the AD neurons, confirmed by histological measurement during the ensuing week (*Figure 2D–F*). Moreover, the AD neurons continued to demonstrate a dramatic increase in sodium current density and evoked action potential frequency compared to WT as they matured in culture, consistent with the notion that these events, reflecting increased electrical excitability, represent a true difference between the PS1 mutant AD and WT neurons (*Figure 3A–E*, *Figure 3—figure supplement 1A–D*).

In further support of these conclusions, the other AD related mutation studied here, mutant APP[swe]/WT AD neurons, did not exhibit larger sodium or potassium current densities, increased synaptic density or electrical capacitance at 2 weeks in culture compared to isogenic WT/WT neurons (*Figure 3A–E*, *Figure 3—figure supplement 2A,B*). The APP[swe]/WT neurons also did not show a difference in synaptic density compared to WT neurons at the 4 and 6 week timepoints (*Figure 3—figure supplement 2A,B*). However, similar to the PS1 mutant neurons, at these later timepoints of 4 and 6 weeks, the APP[swe]/WT neurons showed a decrease in cell capacitance while manifesting a dramatic increase in sodium current density and evoked action potential frequency compared to isogenic WT controls (*Figure 3A–E*), again reflecting their increased electrical excitability. Notably, similar to our observations, previous studies in vivo on APP/PS1 transgenic mice have shown at early stages of AD that neurite loss occurs without change in spine density in surviving dendritic branches compared to WT mice (*Šišková et al., 2014*). Thus, neurite loss and increased excitability occur early in the disease process, prior to the significant neuronal and synaptic loss that occurs subsequently in human AD brains (*DeKosky and Scheff, 1990*; *Terry et al., 1991*), and appear to be faithfully reproduced in our AD hiPSC models.

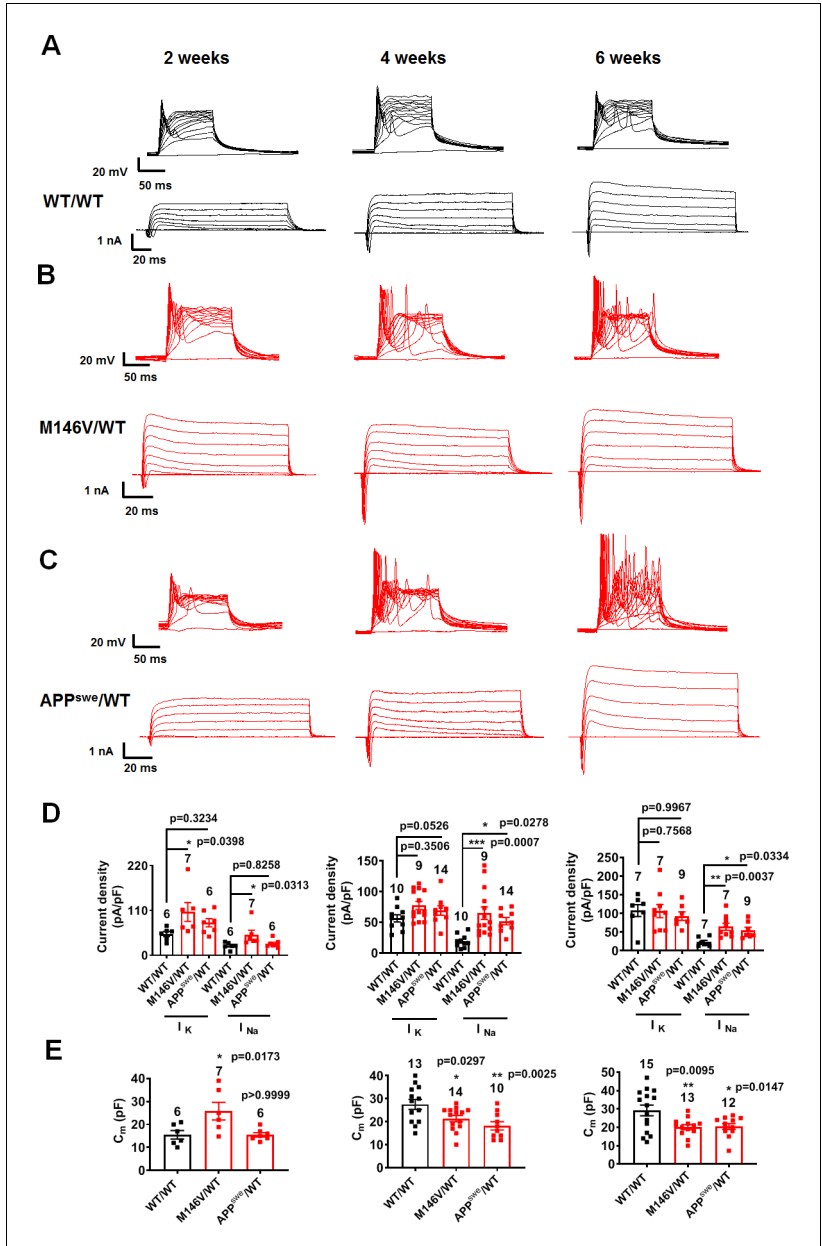

**Figure 3.** Developmental timeline of AD neurons compared to isogenic control neurons. WT/WT hiPSC-derived cerebrocortical neuron data in black, M146V/WT and APP$^{swe}$/WT AD neurons in red. (**A–C**) Representative evoked APs and sodium/potassium currents recorded from WT/WT (**A**), M146V/WT (**B**), and APP$^{swe}$/WT (**C**) hiPSC-derived cerebrocortical neurons in culture for 2 weeks (Left), 4 weeks (Middle) and 6 weeks (Right). (**D**) Sodium (I$_{Na}$) and potassium (I$_K$) current densities. (**E**) Quantification of cell capacitance (C$_m$). Note that potassium current density and cell size were significantly greater in M146V/WT compared to WT/WT at the 2 week timepoint, but at later timepoints there was no difference in potassium current density but the cell capacitance of AD neurons significantly decreased. Data are mean ± SEM. Statistical significance analyzed by ANOVA with post-hoc Dunnett's test.

The online version of this article includes the following source data and figure supplement(s) for figure 3:

**Source data 1.** Excel files containing data shown as summary bar graph in *Figure 3D* (2 weeks, 4 weeks and 6 weeks); *Figure 3E* (2 weeks, 4 weeks and 6 weeks).

**Figure supplement 1.** Developmental timeline of ΔE9/WT neurons compared to isogenic control neurons.

**Figure supplement 2.** Synaptic development in hiPSC-derived AD neurons vs WT neurons.

## Increased excitatory synaptic transmission in AD neurons also contributes to hyperexcitability

Concerning synaptic function, although AD neurons compared to isogenic WT controls showed similar synaptic density by 5 weeks in culture, we observed an increase in frequency and amplitude of spontaneous excitatory postsynaptic currents (sEPSCs) in the AD neurons (*Figure 4A–C*, *Figure 4—figure supplement 1A–E*). Based on prior in vivo work on AD transgenic mice with similar findings (*Šišková et al., 2014*), this increase in sEPSCs could result from improved synaptic efficacy associated with shorter neurites. Additionally, we found an increase in the frequency of miniature EPSCs (mEPSCs) in AD neurons compared to gene-corrected isogenic WT (*Figure 4D,F*, *Figure 4—figure supplement 1F,H*). There was also a statistically significant increase in the amplitude of mEPSCs in ΔE9/WT neurons vs. isogenic WT/WT neurons (*Figure 4—figure supplement 1F,G*), but only a trend in M146V/WT and APP$^{swe}$/WT neurons that did not reach significance (*Figure 4D,E*). An increase in

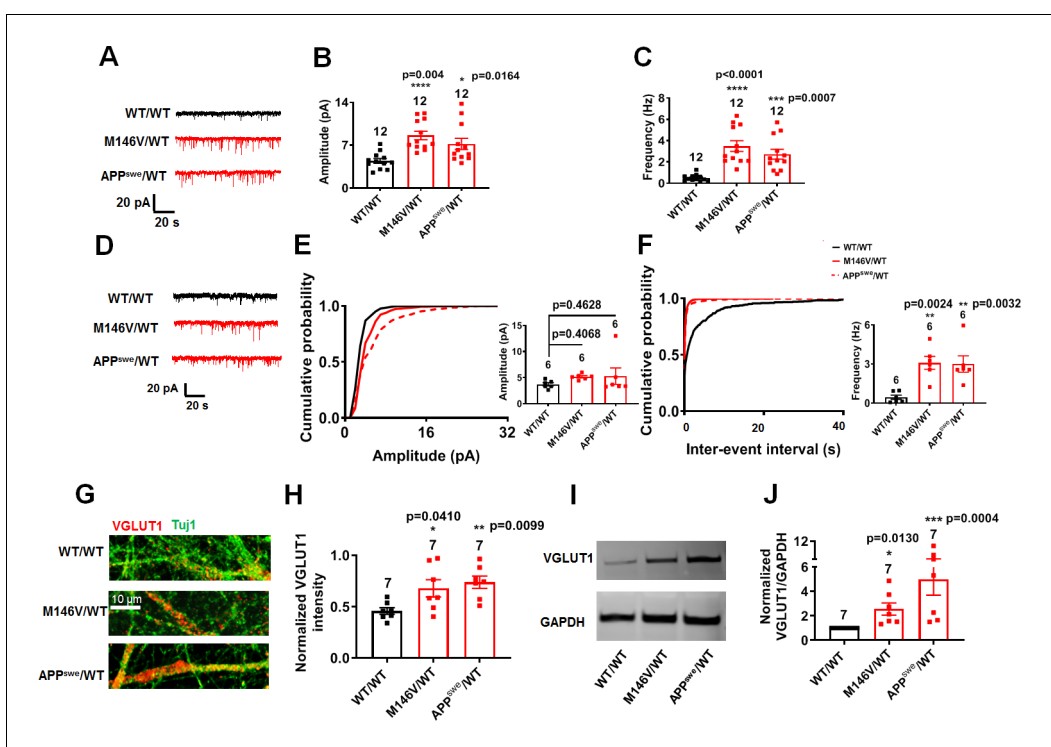

**Figure 4.** AD neurons show disrupted excitatory synaptic transmission compared to isogenic control neurons. (A) Representative spontaneous excitatory postsynaptic currents (sEPSC) recorded at −70 mV from WT/WT, M146V/WT and APP$^{swe}$/WT hiPSC-derived cerebrocortical neurons in culture for 5 weeks. (B,C) Quantification of sEPSC parameters. Quantification of mean amplitude (B) and quantification of mean frequency (C). (D) Representative miniature excitatory postsynaptic currents (mEPSCs) recorded at −70 mV from WT/WT, M146V/WT and APP$^{swe}$/WT hiPSC-derived cerebrocortical neurons at 5 weeks in culture. (E,F) Cumulative probability of mEPSC amplitude (inset: quantification of mean amplitude, E) and mEPSC inter-event interval (inset: quantification of mean frequency, -F). (G) Representative images of VGLUT1 immunostaining. (H) Quantification of VGLUT1 intensity normalized to Tuj1 intensity. Total number of random fields of neuronal cultures analyzed in three separate experiments is listed above the bars. (I) Representative western blot images showing VGLUT1 protein levels. (J) Ratio of VGLUT1/GAPDH normalized to the value of WT VGLUT1/GAPDH. Data are mean ± SEM. Statistical significance analyzed by ANOVA with post-hoc Dunnett's test, or, for immunoblot analysis, by a Kruskal Wallis test followed by Dunn's multiple comparisons (*Deyts et al., 2016*). Number of independent experiments listed above bars.

The online version of this article includes the following source data and figure supplement(s) for figure 4:

**Source data 1.** Excel files containing data shown as summary bar graph in *Figure 4B,C*.
**Figure supplement 1.** ΔE9/WT neurons show increased excitatory synaptic activity compared to isogenic control neurons.
**Figure supplement 2.** Depletion of readily releasable pool in AD and WT neurons.

mEPSC frequency may reflect an increase in synapse number or a presynaptic mechanism, such as an increase in release probability. Along these lines, we found an increase in the level of vesicular glutamate transporter (VGLUT1) in AD cultures compared to WT by immunocytochemistry and western blotting (*Figure 4G–J*, *Figure 4—figure supplement 1I–L*). Since we had found no significant differences between AD and WT neurons with regard to synaptic density at this stage of the disease process, it is therefore likely that presynaptic release was affected in AD. Consistent with this explanation, prior work in rodent hippocampal neurons had shown that Aβ peptide enhances release probability (*Abramov et al., 2009*). To test this possibility more directly, we applied hypertonic sucrose solution to measure the readily releasable pool (RRP) size and found a decrease in RRP, which can be attributed to increased basal release of vesicles from AD neurons (*Figure 4—figure supplement 2A,B*).

## AD neurons manifest impaired inhibition compared to WT neurons

With regard to inhibitory interneurons, among our AD and WT hiPSC-derived cortical neurons, we found γ-aminobutyric acid (GABA)-positive staining in 8–15% and parvalbumin (PV)-positive staining in 3–6% of the cells. Notably, there were significantly fewer inhibitory GABA- and PV-positive neurons in AD cultures compared to WT cultures (*Figure 5—figure supplement 1A–D*). In patch-clamp recordings, we also observed a significant decrease in the frequency of spontaneous and miniature inhibitory postsynaptic currents (sIPSCs, mIPSCs) in AD neurons compared to WT but no significant change in amplitude (*Figure 5A–F*, *Figure 5—figure supplement 2A–F*). Validating these findings in hiPSC-based cultures, a similar decrease in inhibitory neurotransmission was previously reported in vivo in AD transgenic mice (*Verret et al., 2012*). Additionally, we found a decrease in vesicular GABA transporter (VGAT) levels in AD compared to WT cultures, consistent with the notion that a decrease in presynaptic release probability could cause the decrease in mIPSC frequency (*Figure 5G–J*, *Figure 5—figure supplement 2G–J*). Taken together, the increase in excitation and decrease in inhibition that we observed in AD hiPSC-derived neuronal cultures compared to WT controls could contribute to network imbalance between firing stability and synaptic plasticity that has been proposed to contribute to early AD based on data from both human brain and transgenic animal models (*Styr and Slutsky, 2018*).

## γ-Secretase or BACE1 inhibition reverses the hyperexcitability of AD neurons

Next, we further investigated the mechanism underlying the observed hyperactivity in hiPSC-derived AD neurons. Along these lines, we found that incubation in the γ-secretase inhibitor compound E (1 μM) reversed the increase in spontaneous action potential frequency, sodium current density, and excitatory postsynaptic currents in PS1 mutant (M146V/WT) AD neurons (*Figure 6A–G*). Similarly, APP mutant AD neurons (APP^swe^/WT), which bear a mutation adjacent to the BACE1 cleavage site of APP, exhibited a similar decrease in hyperexcitability following incubation with BACE inhibitor IV (1 μM) (*Figure 6A–G*). Isogenic control neurons (WT/WT) did not show significant changes in the presence of γ-secretase or BACE1 inhibitors (*Figure 6A–G*). These results are consistent with the notion that genetic mutations leading to the accumulation of Aβ oligomers directly contribute to the hyperexcitable phenotype of AD hiPSC-derived neurons.

## AD cerebral organoids show aberrantly increased electrical activity

To examine network phenomena in three-dimensions (3D) with our AD hiPSC-based models, we next studied AD neuronal activity in 2-month-old cerebral organoids prepared from WT/WT, M146V/WT, and APP^swe^/WT hiPSC lines. By this age, the cerebral organoids matured to the stage of forming discrete cortical layers and displayed stable electrophysiological properties in WT as well as PS1 (M146V) and APP (APP^swe^) mutants (*Figure 7A,G,H*) (*Lancaster and Knoblich, 2014*; *Velasco et al., 2019*; *Yoon et al., 2019*). Characterizing the cerebral organoids, we found increased Aβ immunostaining and a statistically significant increase in the ratio of secreted Aβ$_{42}$/Aβ$_{40}$ and total Aβ levels in the PS1- mutant and APP-mutant cerebral organoids, respectively, compared to WT (*Figure 7B–D*), as has been reported previously (*Choi et al., 2014*; *Gonzalez et al., 2018*). Similar to our 2D cultures, we found shorter neurites/dendrites in AD cerebral organoids compared to WT (*Figure 7E,F*). AD cerebral organoids also show increased VGLUT1 and decreased VGAT staining

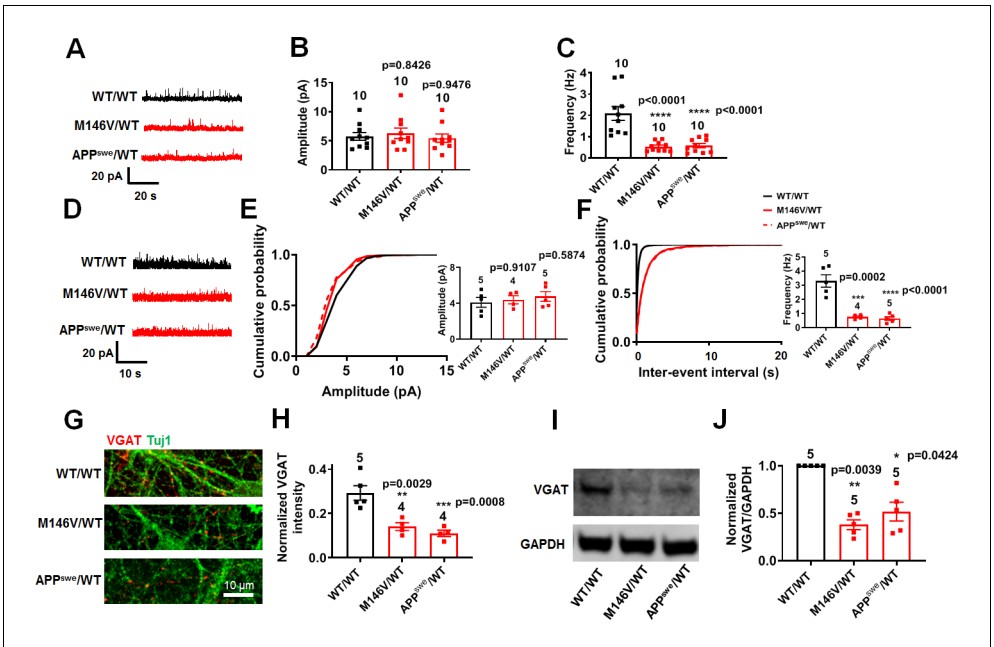

**Figure 5.** AD neurons show diminished inhibitory synaptic transmission compared to isogenic control neurons. WT/WT hiPSC-derived cerebrocortical neuron data in black, M146V/WT and APP^swe/WT in red. (**A**) Representative spontaneous inhibitory postsynaptic currents (sIPSCs) recorded at 0 mV from WT/WT, M146V/WT, and APP^swe/WT hiPSC-derived cerebrocortical neurons in culture for 5 weeks. (**B,C**) Quantification of sIPSC amplitude and frequency. (**D**) Representative miniature inhibitory postsynaptic currents (mIPSCs) recorded at 0 mV from WT/WT, APP^swe/WT, and M146V/WT hiPSC-derived cerebrocortical neurons at 5 weeks in culture. (**E,F**) Cumulative probability of mIPSC amplitude (inset: quantification of mean amplitude) and mIPSC inter-event interval (inset: quantification of mean frequency). (**G**) Representative images of VGAT immunostaining. (**H**) Quantification of VGAT intensity normalized to Tuj1 intensity. Total number of random fields of neuronal cultures analyzed in three separate experiments is listed above the bars. (**I**) Representative western blot images showing expression of VGAT. (**J**) Ratio of VGAT/GAPDH normalized to the value of WT VGAT/GAPDH. Data are mean ± SEM. Statistical significance analyzed by ANOVA with post-hoc Dunnett's test, or by Kruskal Wallis test followed by Dunn's test for western blot quantification). Number of independent experiments listed above bars.

The online version of this article includes the following figure supplement(s) for figure 5:

**Figure supplement 1.** APP^swe/WT and M146V/WT AD cultures contain decreased number of inhibitory neurons compared to WT/WT cultures.

**Figure supplement 2.** ΔE9/WT neurons show decreased inhibitory synaptic activity compared to isogenic control neurons.

(*Figure 7—figure supplement 1A,B*). Notably, AD cerebral organoids plated in a multielectrode array (MEA) recording chamber displayed a significant increase in action potential firing rate compared to WT (*Figure 7G,H*), mimicking their 2D counterparts.

## Discussion

### hiPSC-derived neurons in 2D cultures and 3D organoids as a model system for AD

While no in vitro system can be expected to simulate every aspect of a complex neurodegenerative disorder of aging such as AD, our results show that hiPSC models derived from AD patient mutations may prove useful for two reasons. First, AD hiPSCs allow us to study aspects of the electrical properties of AD neurons in a human context in both 2D cultures and 3D cerebral organoids and compare them to known in vivo phenotypic studies in AD transgenic mouse models and in human AD brain. Indeed, we found that hiPSC-derived neurons in 2D culture differentiated for ≥5 weeks displayed stable phenotypes that resemble several aspects of the electrophysiological abnormalities,

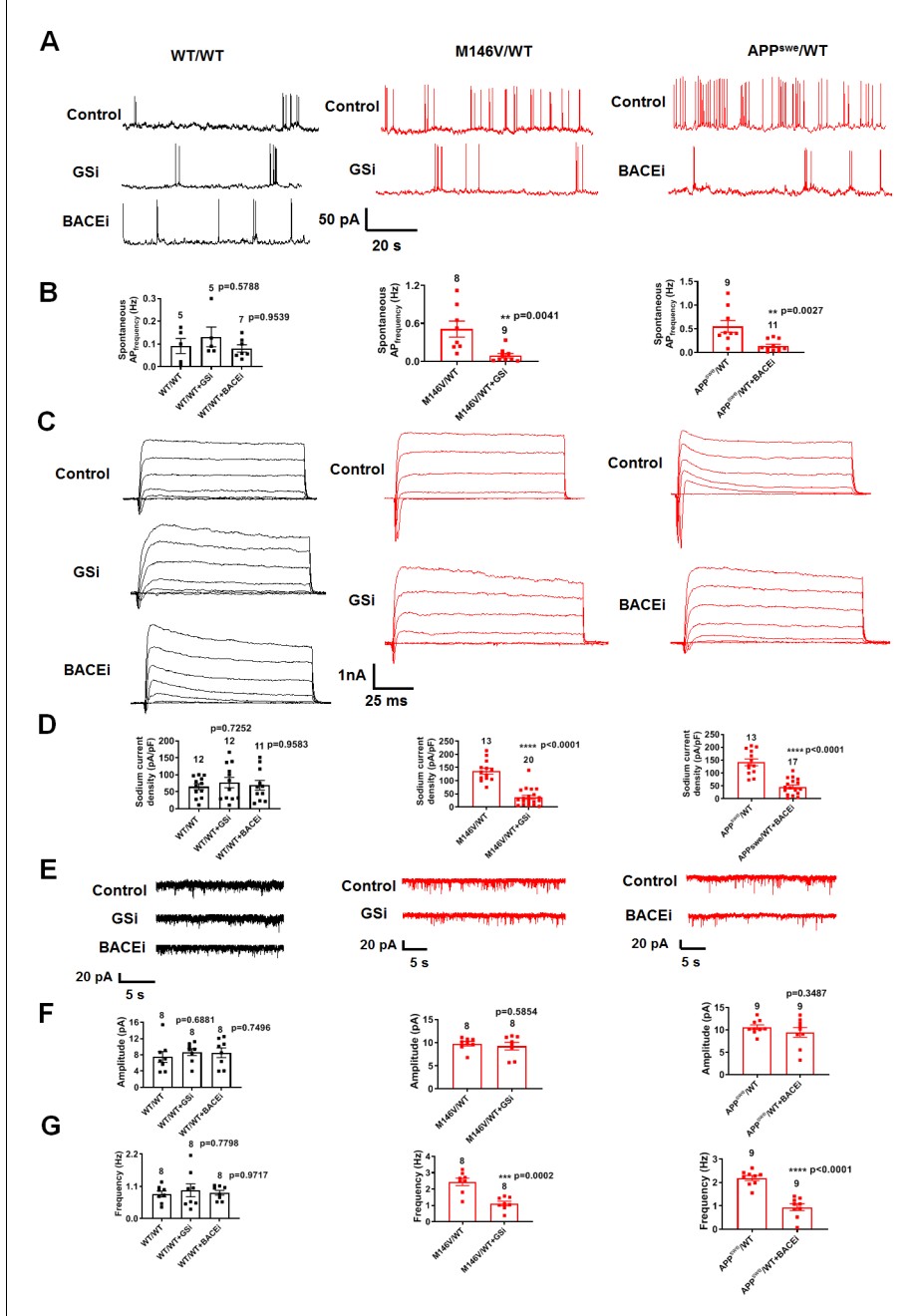

**Figure 6.** γ-Secretase inhibitor or BACE1 inhibitor prevent the hyperexcitability of AD neurons. (**A**) Spontaneous action potentials (sAP) at resting membrane potential (RMP) in the presence of γ-secretase inhibitor (GSi) for 2 days or BACE1 inhibitor (BACEi) for 4 days. WT/WT (*left*, in black) hiPSC-derived cerebrocortical neuron data, M146V/WT (center, in red), and APP^swe/WT (right, in red). (**B**) Quantification of sAP frequency. (**C**) Representative sodium and potassium currents recorded from neurons clamped at −70 mV. (**D**) Current densities. (**E**) Representative spontaneous excitatory postsynaptic currents (sEPSCs) recorded at −70 mV from WT/WT and ΔE9/WT hiPSC-derived cerebrocortical neurons in culture for 5 weeks. (**F,G**) Quantification of sEPSC parameters. Quantification of mean amplitude (**F**) and quantification of mean frequency (**G**). Data are mean ± SEM. Statistical significance analyzed by ANOVA with post hoc Dunnett's test for multiple comparisons and by Student's t test for comparison between two groups.

The online version of this article includes the following source data for figure 6:

**Source data 1.** Excel files containing data shown as summary bar graph in *Figure 6B* (WT/WT, M146V/WT, APP^swe/WT); *Figure 6D* (WT/WT, M146V/WT, APP^swe/WT).

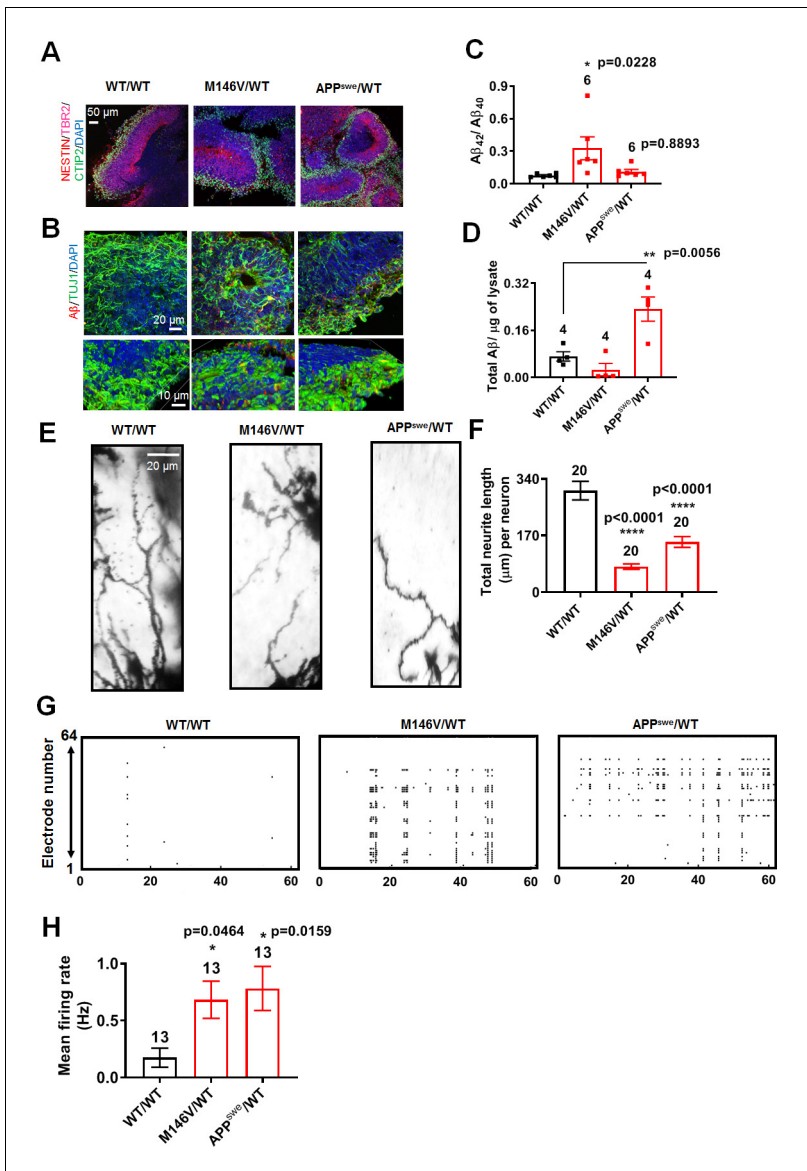

**Figure 7.** AD cerebral organoids show increased synchronous burst activity compared to WT. (**A**) Representative immunostaining images of Nestin, TBR2, and CTIP2 in 2-month-old WT/WT, M146V/WT, and APP^swe^/WT organoids showing cortical layer formation. (**B**) Representative Aβ immunostaining in WT/WT, M146V/WT, and APP^swe^/WT cerebral organoids; lower panel shows 3D projection of Aβ immunostaining. Note increased Aβ deposition in 2-month-old AD organoids. (**C**) Ratio of $A\beta_{42}/A\beta_{40}$ quantified by ELISA from cultures at 2 months. (**D**) Quantification of Aβ levels normalized to total protein from lysates at 2 months. (**E**) Golgi staining of neurons in 200 µm sections of organoids. (**F**) Quantification of total dendrite length. Total number of neurons analyzed in 3–4 separate experiments is listed above the bars. (**G**) Representative raster plots of MEA recordings in WT/WT, M146V/WT, and APP^swe^/WT cerebral organoids at 2 months. (**H**) Quantification of mean firing rate. Data are mean ± SEM. Statistical significance analyzed by ANOVA with post-hoc Dunnett's test. Number of independent experiments listed above the bars.

The online version of this article includes the following source data and figure supplement(s) for figure 7:

**Source data 1.** Excel files containing data shown as summary bar graph in *Figure 7F,H*.

**Figure supplement 1.** AD cerebral organoids show increased VGLUT1 levels and decreased VGAT levels compared to WT.

including hyperexcitability, previously observed in vivo in human AD brain and transgenic mouse models. Moreover, after >2 months of maturation, cerebral organoids showed clear layer formation, allowing us to examine early neural network aspects of the disease in cells that bear familial AD-causing mutations. Second, given that similar electrical events are observed in the hiPSC system as in in vivo models, this reductionist approach allows us to begin to elucidate the underlying mechanism(s) for the aberrant electrical activity in AD brains.

Since familial AD generally does not manifest symptoms until 40–50 years of age (*Kwart et al., 2019*), it is somewhat surprising that our hiPSC-based models display electrophysiological features of the disease phenotype. This is especially perplexing because hiPSCs are reprogrammed to an earlier epigenetic state and hence would lack age-related changes (*Mertens et al., 2018*; *Miller et al., 2013*). However, in the intact human brain there are homeostatic mechanisms that may play a role in delaying the onset of the disease, and these 'brakes' on the system are not present in our simplified culture systems. While no AD animal model or culture system can recapitulate factors contributing to human aging (*LaFerla and Green, 2012*), genetic factors contributing to AD, as studied here, may be reproduced in a hiPSC model. For example, our study and others have shown that hiPSC-derived neurons with early onset AD mutations manifest greater Aβ secretion and deposition early on in their development in culture (*Kwart et al., 2019*; *Paquet et al., 2016*; *Woodruff et al., 2013*). These early changes indicate that early stages of AD pathophysiology do indeed occur in these in vitro systems (*Penney et al., 2019*). Moreover, 3D cerebral organoids derived from these hiPSCs display accumulation of Aβ over several months and can be maintained for over a year, which is still far from the timescale of human brain aging, but comparable to many AD animal models (*Choi et al., 2014*; *Gonzalez et al., 2018*). Thus, although hiPSC models of AD are far from perfect, they are reproducible, allow for rigorous investigation of underlying mechanisms for specific phenotypes found in human AD brains such as hyperexcitability, and thus afford the opportunity of elucidating novel insight into early stages of human AD pathophysiology.

In particular, we were interested in studying excitability/firing properties and synaptic dysfunction that might precede and predispose the synaptic loss that represents the hallmark of advanced stages of symptomatic AD (*DeKosky and Scheff, 1990*; *Terry et al., 1991*). Thus, our findings potentially yield insight into the origin of the observed hyperexcitable phenotype that occurs in human AD patient brains and AD transgenic mouse models as well as in our hiPSC model systems.

## Aberrant activity in AD neurons associated with morphological changes and ion channel dysfunction

Hyperexcitability and ion channel dysfunction have been implicated in the pathophysiology of AD in prior studies in a variety of animal and cell-based models (*Kim et al., 2007*; *Liu et al., 2015*; *Palop and Mucke, 2016*; *Talantova et al., 2013*). However, the mechanism underlying this hyperexcitability and its effect on network signaling has not been fully elucidated. While inhibitory neuron dysfunction may be essential for aberrant excitability in AD brain (see, e.g., *Verret et al., 2012*), we also found a contribution of aberrant excitatory neuronal activity in our AD hiPSC-derived neurons, as suggested previously in AD transgenic mouse models (*Šišková et al., 2014*). Specifically, comparing AD hiPSCs and isogenic WT controls, we found that AD neurons exhibited a decrease in neurite length. As shown previously in APP/PS1 AD transgenic mice, such loss of dendrites can contribute to abnormal postsynaptic integration of currents, leading to abnormal hypersynchronous network activity (*Šišková et al., 2014*). Our results showing hyperactivity in AD hiPSC-derived neuronal cultures and cerebral organoids coupled with decreased neurite length are in close agreement with prior results in the AD transgenic mouse model, and thus validate our in vitro human model with in vivo findings.

Additionally, we found an increase in sodium current density and altered kinetics that may contribute to the increase in excitability observed in human AD brains, for example as recorded on EEG. While previous reports had shown a decrease in sodium current density in AD neurons in cell-based and transgenic models (*Kim et al., 2007*; *Palop and Mucke, 2016*), this decrease was observed in PV-positive inhibitory interneurons, rather than in excitatory neurons as found here. However, in support of our findings, enhanced Nav1.6 sodium channel expression was recently demonstrated in APP/PS1 bigenic AD mice, suggesting that Aβ may be responsible for the resulting increase in voltage-gated sodium current in excitatory neurons (*Liu et al., 2015*; *Wang et al., 2016*).

## Accelerated maturation in PS1 mutant AD hiPSC-derived neurons compared to isogenic WT controls

Interestingly, we also observed developmental differences between PS1 (but not hAPP) mutant AD neurons and isogenic WT controls. For example, PS1 mutant AD hiPSC-derived neurons manifested action potentials and larger neuronal size at 2 weeks of culture. Based on prior evidence, the accelerated maturation of PS1 mutant neurons may occur because of decreased γ-secretase activity, leading to reduced Notch signaling, which would otherwise inhibit differentiation of neural progenitors (*Li et al., 2011*). In support of this hypothesis, APP mutant AD neurons, which display normal γ-secretase activity, did not exhibit more rapid maturation compared to isogenic WT controls. Moreover, by week four in culture, WT neurons had caught up with the PS1 mutant neurons with regard to the various neuronal parameters. Indeed, both PS1 and APP mutant AD neurons showed a decrease in cell capacitance at the later timepoints, reflecting neuritic/dendritic loss compared to WT. Nonetheless, both PS1 and mutant APP AD neurons maintained their hyperactivity throughout the culture period ($\geq$7 weeks), suggesting that hyperactivity represents a true pathophysiological phenotype that begins to manifest early in the disease process.

## Synaptic dysfunction contributes to aberrant activity in AD neurons

In our AD neurons we also observed an increase in sEPSCs, which may represent more effective integration of synaptic inputs. This synaptic facilitation may arise in part from the observed decrease in neurite length and branching in the AD neurons. In a similar fashion, in an AD transgenic mouse model, Stefan Remy's group has shown that dendritic degeneration resulted in enhanced synaptic efficacy and increased spontaneous excitatory postsynaptic potential (EPSP) amplitude and frequency (*Šišková et al., 2014*).

We also observed an increase in the frequency of mEPSCs in electrophysiological recordings and in VGLUT1 levels histologically in the AD hiPSC-derived excitatory neurons, consistent with the notion of increased presynaptic release probability. In accord with these findings, previous reports have shown that Aβ peptide increases release probability at excitatory synapses (*Abramov et al., 2009*; *Fogel et al., 2014*; *Parodi et al., 2010*; *Wang et al., 2017*). Moreover, the increased levels of VGLUT1 found in our study are similar to those shown in human AD brain tissue and mouse models (*Sokolow et al., 2012*; *Timmer et al., 2014*). Collectively, these findings suggest that both morphological changes in neurites and increased release probability may contribute to excitatory synaptic hyperactivity.

In addition to an increase in excitatory neurotransmission, we found a decrease in inhibitory activity, as reflected by decrements in sIPSC and mIPSC frequency in the AD neuronal cultures compared to WT. This could be in part due to a decrease in the number of GABAergic neurons and thus inhibitory endings. In fact, we found a decrease in VGAT levels, as observed previously in AD transgenic mouse models (*Fuhrer et al., 2017*). Additionally, prior reports have implicated a decrease in PV interneuron activity because of decreased sodium channel expression in AD transgenic mouse models (*Palop and Mucke, 2016*). In agreement with our finding of decreased inhibitory neuronal numbers in AD hiPSC-derived cultures, a decrease in the interneuron population has also been suggested as a contributory factor to neuronal hyperactivity based on AD mouse models (*Schmid et al., 2016*).

We also observed that AD hiPSC-derived neuronal hyperactivity could be significantly ameliorated by γ-secretase or BACE1 inhibitors, suggesting that aberrant APP processing contributes to the observed pathophysiology. Along these lines, multiple pathological species formed from abnormal APP processing have been reported, including, for example, Aβ oligomers, APP intracellular domain (AICD), and the β-secretase-derived fragment, C99 (β-CTF) (*Ghosal et al., 2009*; *Kwart et al., 2019*). Several studies including our own have suggested that Aβ dimers or oligomers may contribute to hyperexcitability in AD (*Talantova et al., 2013*; *Selkoe, 2019*; *Zott et al., 2019*). Our new findings yield mechanistic insight into these observations.

In conclusion, mechanisms affecting both excitatory and inhibitory neurons can contribute to the observed hyperactivity in AD neuronal networks. Here we have used AD hiPSC-derived neurons to show how dysfunction emanating from degenerating neurite morphology, ion current density, firing properties, and altered synaptic transmission can contribute to the aberrant hyperactivity of AD neuronal networks.

# Materials and methods

## Key resources table

| Reagents or resource | Designation | Source or reference | Identifiers | Additional information |
|---|---|---|---|---|
| Cell culture | Matrigel | Corning | Cat # 354230 | |
| Cell culture | mTeSR 1 | STEMCELL Technologies | Cat # 85850 | |
| Cell culture | Dorsomorphin | Tocris | Cat # 3093 | |
| Cell culture | A83-01 | Stemgent | Cat # 04–0014 | |
| Cell culture | PNU74654 | Tocris | Cat # 3534 | |
| Cell culture | DMEM/F12 | ThermoFisher | Cat # 10565018 | |
| Cell culture | N2 | ThermoFisher | Cat # 17502048 | |
| Cell culture | B27 | ThermoFisher | Cat # 17504044 | |
| Cell culture | FGF | R and D | Cat # 4114-TC | |
| Cell culture | p-ornithine | Millipore Sigma | Cat # 27378490 | |
| Cell culture | laminin | Trevigen | Cat # 340001001 | |
| Cell culture | Knockout Serum Replacement | ThermoFisher | Cat # 10828028 | |
| Cell culture | β-Mercaptoethanol | ThermoFisher | Cat # 21985023 | |
| Cell culture | BDNF | Peprotech | Cat # AF-450–02 | |
| Cell culture | GDNF | Peprotech | Cat # AF-450–10 | |
| Cell culture | ES-FBS | ThermoFisher | Cat # 16141079 | |
| Organoid culture | Cerebral Organoid Kit | STEMCELL Technologies | Cat # 08570 | |
| Critical commercial assay | V-PLEX Plus Aβ Peptide Panel 1 (6E10) Kit | Meso Scale Discovery | Cat # K15200 | |
| Electrophysiology | K-Gluconate | Millipore Sigma | Cat # P1847 | |
| Electrophysiology | Cs-Gluconate | Hello Bio | Cat # HB4822 | |
| Electrophysiology | KCl | Millipore Sigma | Cat # 60128 | |
| Electrophysiology | CsCl | Millipore Sigma | Cat # C4036 | |
| Electrophysiology | $MgCl_2$ | Fluka | Cat # 63020 | |
| Electrophysiology | HEPES | Millipore Sigma | Cat # H4034 | |
| Electrophysiology | EGTA | Millipore Sigma | Cat # 3889 | |
| Electrophysiology | Mg-ATP | Millipore Sigma | Cat # A9187 | |
| Electrophysiology | HBSS | ThermoFisher | Cat # 14175079 | |
| Electrophysiology | $CaCl_2$ | Millipore Sigma | Cat # C5080 | |
| Electrophysiology | Glycine | Millipore Sigma | Cat # G2879 | |
| Electrophysiology | Tetrodotoxin citrate | Hello Bio | Cat # HB 1035 | |
| Electrophysiology | β-Secretase Inhibitor IV | EMD Millipore | Cat # 565788 | |

*Continued on next page*

*Continued*

| Reagents or resource | Designation | Source or reference | Identifiers | Additional information |
|---|---|---|---|---|
| Electrophysiology | Compound E | EMD Millipore | Cat # 565790 | |
| Immunocytochemistry (ICC) | PFA | Alfa Aesar | Cat # 43368 | |
| ICC | BSA | Millipore Sigma | Cat # A0336 | |
| ICC | Triton X-100 | Millipore Sigma | Cat # T8532 | |
| ICC antibody | DAPI | Invitrogen | Cat # D1306; RRID:AB_2629482 | 1:500 |
| ICC antibody | Chicken polyclonal Anti-Tuj1 | Abcam | Cat # ab41489; RRID:AB_727049 | 1:200 |
| ICC antibody | Mouse monoclonal Anti-Synapsin 1 | Synaptic Systems | Cat # 106 011; RRID:AB_2619772 | 1:500 |
| ICC antibody | Rabbit polyclonal Anti-Homer 1 | Synaptic Systems | Cat # 160 003; RRID:AB_887730 | 1:200 |
| ICC antibody | Mouse monoclonal Anti-VGAT | Synaptic Systems | Cat # 131 011; RRID:AB_887868 | 1:200 |
| ICC antibody | Rabbit polyclonal Anti-VGLUT1 | EMD Millipore | Cat # ABN1647; RRID:AB_2814811 | 1:500 |
| ICC antibody | Rabbit polyclonal Anti-GABA | Sigma | Cat # A2052; RRID:AB_477652 | 1:100 |
| ICC antibody | Mouse monoclonal Anti-Parvalbumin | EMD Millipore | Cat # MAB1572; RRID:AB_2174013 | 1:100 |
| ICC antibody | Rabbit polyclonal Anti-FOXG1 | Abcam | Cat # ab18259; RRID:AB_732415 | 1:250 |
| ICC antibody | Rat monoclonal Anti-CTIP2 | Abcam | Cat # ab18465; RRID:AB_2064130 | 1:200 |
| ICC antibody | Rabbit polyclonal Anti-TBR2 | Abcam | Cat # ab23345; RRID:AB_778267 | 1:300 |
| ICC antibody | Mouse monoclonal Anti-Nestin | Abcam | Cat # ab22035; RRID:AB_446723 | 1:200 |
| ICC antibody | Anti-Amyloid $\beta$ 1–16 | Biolegend | Cat # 803001; RRID:AB_2564653 | 1:2000 |
| Immunoblot antibody | Mouse monoclonal Anti-VGAT | Synaptic Systems | Cat # 131011; RRID:AB_887868 | 1:500 |
| Immunoblot antibody | Rabbit polyclonal Anti-VGLUT1 | Synaptic Systems | Cat # 135303; RRID:AB_887875 | 1:1000 |
| Software and Algorithms | Clampex v10.6 (pClamp) | Molecular Devices | RRID:SCR_011323 | |
| Software and Algorithms | Clampfit v10.6 (pClamp) | Molecular Devices | RRID:SCR_011323 | |
| Software and Algorithms | Mini Analysis | Synapstosoft | http://www.synaptosoft.com/MiniAnalysis/; RRID:SCR_002184 | |
| Software and Algorithms | ImageJ | NIH | https://imagej.nih.gov/ij/; RRID:SCR_003070 | |
| Software and Algorithms | Prism7 | GraphPad | http://www.graphpad.com; RRID:SCR_002798 | |

## hiPSC lines

Four hiPSC lines were used in this study:

 a. ΔE9/WT-hiPSC line bearing the PSEN1 ΔE9 mutation and isogenic WT control (from the Lawrence Goldstein Lab, University of California, San Diego).

b. M146V/WT and APP^swe/WT hiPSC lines bearing the PSEN1 M146V and APP Swedish muta-tion, respectively, and the associated isogenic WT control (from the Marc Tessier-Lavigne lab, Rockefeller University/Stanford University).

The details regarding these lines have been previously published (*Paquet et al., 2016*; *Woodruff et al., 2013*).

## hiPSC maintenance and differentiation

Differentiation of hiPSCs was performed using standard protocols for generating cerebrocortical neurons (*Talantova et al., 2013*). Briefly, feeder-free hiPSCs were cultured on Matrigel in mTeSR1 medium (StemCell Technologies, Vancouver, Canada). Differentiation was induced by exposure to a cocktail of small molecules containing 2 µM each of Dorsomorphin (bone morphogenetic protein inhibitor, Tocris, Minneapolis, MN), A83-01 (Activin/Nodal inhibitor, Tocris) and PNU74654 (Wnt/β-catenin inhibitor, Tocris) for 6 days in DMEM/F12 medium supplemented with 20% Knock Out Serum Replacement (Invitrogen, Carlsbad, CA). Cells were scraped manually to form PAX6$^+$ neurospheres, which were maintained for about 2 weeks in DMEM/F12 medium supplemented with N2 and B27 (Invitrogen) and 20 ng ml$^{-1}$ of basic FGF. The neurospheres were subsequently seeded on p-orni-thine/laminin-coated dishes to form a monolayer of human neural progenitor cells (hNPCs) contain-ing rosettes, which were manually picked and expanded.

For terminal differentiation, two different protocols were used to confirm that the results obtained did not vary with differentiation protocol. Similar data were obtained with each of the fol-lowing protocols: (i) a 1:1 ratio of hNPCs and neonatal mouse astrocytes were seeded onto p-orni-thine/laminin-coated glass coverslips ($7 \times 10^5$ cells cm$^{-2}$) in DMEM/F12 medium supplemented with N2 and B27, brain derived neurotrophic factor (20 ng ml$^{-1}$), glial cell line-derived neurotrophic fac-tor (20 ng ml$^{-1}$), and 0.5% FBS. Alternatively, (ii) hNPCs were treated with 100 nM compound E (EMD Millipore, Temecula, CA) in BrainPhys medium (StemCell Technologies) for 48 hr and then maintained in culture in BrainPhys medium. Note that for experiments in which synapse formation was analyzed, the hiPSC-derived neurons were plated on a bed of astrocytes to foster synapse for-mation (*Ullian et al., 2001*). Both of our differentiation protocols produced 8–15% inhibitory neu-rons, as monitored by immunocytochemistry with anti-GABA antibody (Sigma, St. Louis, MO). Cells at week 3 of terminal differentiation were switched to BrainPhys medium, and most experiments were conducted after 5–6 weeks of differentiation.

One NPC line per genotype was isolated and DNA sequencing to confirm the mutations was done using the primers supplied by the original lab (*Paquet et al., 2016*).

## Cerebral organoid culture preparation

Differentiation of hiPSCs to generate cerebral organoids was performed using an organoid differen-tiation kit (StemCell Technologies). Briefly, hiPSCs were seeded in 96-well plates with embryoid body (EB) seeding media and maintained with EB formation media for 5 days. Thereafter the newly formed EBs were transferred to 24-well plates and maintained in induction medium. After 48 hr., the EBs that had developed translucent edges were embedded in Matrigel and transferred to a 6-well plate containing expansion medium. Three days later, embedded EBs that had developed budding neuroepithelia were considered newly formed organoids. These organoids were then transferred to maturation media for development of cortical-like layers and other more mature phenotypes. All plates used were low-attachment plates, as recommended by the manufacturer (StemCell Technolo-gies). The age of the organoids was determined from the start of the maturation step of the protocol, that is the day they were placed in maturation medium. By 8 weeks of age, we observed the formation of cortical-like layers. Organoids were maintained on an orbital shaker until used in experiments.

## Enzyme-linked immunosorbent assay (ELISA)

After 6 weeks of terminal differentiation, the level of human Aβ peptides 1–38, 1–40 and 1–42 were measured in the culture medium of AD hiPSC-derived neurons by ELISA (V-PLEX Aβ Peptide Panel 1 (6E10) Kit, Meso Scale Discovery). Culture medium was collected for one week after the prior media change and processed as recommended by the manufacturer. The ELISA plate was read in a MESO QuickPlex SQ 120 reader (Meso Scale Discovery). Additionally, we determined the level of Aβ

peptides secreted by 1–2 month-old cerebral organoids; for this purpose, culture medium was collected 3 days after the prior medium change and processed as above. All samples were taken from age-matched organoids and measured in duplicate for statistical analysis.

## Electrophysiology and pharmacology

Whole-cell recordings with a patch electrode were performed using 3- to 5 MΩ resistance pipettes filled with an internal solution composed of (in mM): K-gluconate, 120; KCl, 5; MgCl$_2$, 2; HEPES, 10; EGTA; 10; Mg-ATP, 4; pH 7.4 and mOsm 290. The external solution was composed of Ca$^{2+}$ and Mg$^{2+}$-free Hank's Balanced Salt Solution (HBSS; GIBCO, Gaithersburg, MD) to which we added CaCl$_2$, 2 mM; HEPES, 10 mM; glycine, 20 µM; pH 7.4. Patch pipettes were pulled from borosilicate glass capillaries (G150F-3; Warner Instruments, Hamden, CT) using a micropipette puller (P97; Sutter Instruments, Novato, CA). All recordings were performed using a Multiclamp 700B amplifier (Molecular Devices) at a data sampling rate of 10 kHz with a Digidata 1440A analog-to-digital convertor (Molecular Devices). Voltage-clamp and current-clamp protocols were applied using Clampex v10.6 (Molecular Devices).

Preliminary analysis and offline filtering at 500 Hz were achieved using Clampfit v10.6 (Molecular Devices). Spontaneous postsynaptic currents (sPSCs) were recorded in gap-free mode at a holding potential of −70 mV and 0 mV. The internal solution comprised of (in mM): K-gluconate, 120; KCl, 5; MgCl$_2$, 2; HEPES, 10; EGTA; 10; Mg-ATP, 4; pH 7.4 and mOsm 290. Under these conditions at 21 °C, the chloride ion reversal potential was approximately −70 mV; hence, synaptic currents recorded at −70 mV represented excitatory responses. Since the cation reversal potential was approximately 0 mV, synaptic currents recorded at 0 mV were predominantly inhibitory in nature. To assess the effect of γ-secretase and BACE1 inhibitors, we performed recordings of spontaneous action potentials, voltage gated sodium channel currents, and sEPSCs with the aforementioned external and internal solutions after incubating hiPSC-derived neuronal cultures for 2 days in the γ-secretase inhibitor compound E (EMD Millipore) or 4 days in BACE inhibitor IV (EMD Millipore), respectively, each at 1 µM (*Kwart et al., 2019*). Miniature excitatory postsynaptic currents (mEPSCs) were recorded at −70 mV and miniature inhibitory postsynaptic currents (mIPSCs) were recorded at 0 mV at 21 °C after equilibrium in tetrodotoxin (TTX) for at least 20 min. Internal solution used for recording mEPSCs and mIPSCs comprised of (in mM): Cs-gluconate 130; CsCl 5; MgCl$_2$, 2; HEPES, 10; EGTA; 1; Mg-ATP, 4; pH 7.4 and mOsm 290. The external solution used was the same as that listed above with the addition of 1 µM TTX. To calculate the frequency and amplitude of spontaneous synaptic events, Mini Analysis software (Synapstosoft, Fort Lee, NJ) was used.

To determine the readily releasable pool (RRP) of synaptic vesicles, we used a hypertonic sucrose solution containing 500 mM sucrose and 1 µM TTX (*Cho and Askwith, 2008*). The internal solution used for the RRP experiments was same as that used for recording miniature postsynaptic currents. Sucrose evoked responses were recorded at a holding potential of −70 mV. For estimating the size of the RRP, total charge transferred during the transient phase of the sucrose-mediated response was quantified by calculating the area under the curve.

For measurements of the slope of action potential decay, we quantified the first evoked action potential elicited by a series of current-clamp steps. To measure sodium current rise time and decay time constants, 10–90% of the slope of the first sodium current trace elicited by a voltage-clamp step protocol was used for analysis. All quantifications were performed using Clampfit v10.6 (Molecular devices).

## Immunocytochemistry of 2D neuronal cultures

Cells were fixed with 4% PFA for 20 min, washed with PBS, and blocked with 3% BSA/0.3% Triton X-100 in PBS for 30 min. Cells were incubated with primary antibody overnight, and the appropriate Alexa Fluor (488, 555, 647) conjugated secondary antibody was used at 1:1000 dilution. Primary antibodies and dilutions were as follows: Tuj1 (1:200; Abcam, Cambridge, MA), FOXG1 (1:250; Abcam), Synapsin 1 (1:500; Synaptic Systems, Gottingen, Germany), Homer (1:200; Synaptic Systems), VGAT (1:200, Synaptic Systems), VGLUT1 (1:500, EMD Millipore), GABA (1:100, Sigma, St. Louis, MO), Parvalbumin (1:100, EMD Millipore). Cells were counterstained with DAPI (1:500; Invitrogen).

Images were collected by laser scanning confocal fluorescence microscopy (Nikon A1) by an observer blinded to experimental group. The area covered by Tuj1 immunostaining (imaged using a

20X objective) was analyzed with ImageJ 1.52 p. Somal area was determined manually by marking the Tuj1-positive area around DAPI-stained nuclei using the freehand selection tool in ImageJ. Neurite area was determined by subtracting the somal area from the total Tuj1-immunostained area. For synapse quantification, colocalized presynaptic (Synapsin 1) and postsynaptic (Homer) puncta within Tuj1-expressing neuronal cells (imaged using a 60X objective) were counted using ImageJ. The number of synaptic puncta were scored per μm of neurite length. For VGLUT1 and VGAT immunofluorescence images (also collected using a 60X objective), total VGLUT1 and VGAT intensity was quantified using ImageJ and normalized to total Tuj1 intensity. All analyses were performed by an observer masked to experimental condition.

## Immunocytochemistry of cerebral organoids

After 2 months of development, cerebral organoids were fixed in 4% PFA at 4 ˚C overnight followed by serial incubation in 15% and 30% sucrose in PBS overnight. Fixed organoids were then placed in a cryomold and embedded in tissue freezing medium (General Data, Cincinnati, OH). Thereafter, they were flash frozen with isopentane and liquid nitrogen and stored at −80 ˚C. Frozen organoids were mounted on sample stubs of a cryostat using optimal cutting temperature (OCT) compound and then sectioned at 20 μm thickness. The sections were collected on glass slides and stained with antibodies to CTIP2 (1:200; Abcam), Tbr2 (1:300; Abcam), Nestin (1:200; Abcam), Tuj1 (1:200; Abcam), VGAT (1:200, Synaptic Systems), VGLUT1 (1:500, EMD Millipore), or Aβ peptide (1:2000; Biolegend). Sections were then imaged with a laser scanning confocal fluorescence microscope (Nikon A1).

## Transfection and neurite tracing

Neurons differentiated from hNPCs via incubation with Compound E were transfected with pCAG-MaxGFP (GFP) using lipofectamine 2000 (Invitrogen). GFP-labeled neurons were imaged with a 10X objective using a Nikon A1 confocal microscope. The imaged neurons were then traced using the neurite tracer plugin available in Image J. An observer masked to the experimental group traced the neurites for subsequent quantification.

## Immunoblots

Western blots were performed to assess and quantify VGLUT1 and VGAT levels. For this purpose, 4–5 week-old neuronal cultures were washed with PBS and lysed in RIPA lysis buffer to which Halt Protease Inhibitor Cocktail was added (Thermo Fisher Scientific). Lysates diluted in SDS sample buffer plus β-mercaptoethanol were separated on blot 4–12% Bis Tris Plus gels (Invitrogen) and transferred to PVDF membrane (EMD Millipore). Primary antibodies used included anti-VGAT antibody (1:500 dilution, Synaptic Systems) and anti-VGLUT1 antibody (1:1000 dilution, Synaptic Systems). Secondary antibodies were purchased from LI-COR Biosciences. Labeled proteins were detected using an Odyssey Infrared Imaging System (LI-COR Biosciences). Densitometry of protein band was analyzed with ImageJ software.

## Golgi staining

Golgi staining was performed using a sliceGolgi Kit (Cat#003760, Bioenno Tech, LLC). Organoids were fixed in the aldehyde fixative provided by the manufacturer overnight at 4 ˚C. The organoids were then embedded in 4% low melting agarose in high sucrose cutting solution and left to solidify. The organoids were sectioned at 200 μm thickness using a vibratome (VT 1000S, Leica Biosystems). The sections were collected in 6-well plates and immersed in impregnation solution for 7 days. The sections were further processed as recommended by the manufacturer. After mounting with permount (Fisher Chemicals), the sections were imaged in bright-field mode. Neurons were then traced using ImageJ and analyzed for neurite length by an observer masked to experimental group.

## MEA recordings

For MEA recordings, 6-week-old cerebral organoids were plated on CytoView 12-well plates (Axion Biosystems) coated with 0.1% polyethyleneimine (PEI) and laminin (10 μg/ml). Spontaneous electrical activity was recorded two weeks later. Recordings were performed at 37 ˚C using the standard

neural settings in 'neural spikes analog mode' (Maestro Axis Software, version 2.4.2, Axion Biosystems). Analysis was performed using the neural metric tool.

## Data analysis and statistics

Data are presented as mean ± SEM. Statistical analyses were performed using GraphPad Prism software. In general, for single comparisons, a two-tailed unpaired Student's t-test was used for determining statistical significance. For multiple comparisons, a two-tailed ANOVA followed by a post-hoc test was utilized. For non-parametric data, single comparisons were made with a two-sided Mann-Whitney U test, and for multiple comparisons, a Kruskal Wallis test was used followed by a post-hoc test. A p value < 0.05 was considered to be statistically significant.

## Experimental design

Inclusion and exclusion criteria of any data or subjects: We did not exclude any data from analysis. All data presented in the figures were biological replicates as they were from separate samples and are listed in the figures as sample size. In ELISA experiments duplicates of each sample, considered as technical replicates, were used and the mean value for each sample was taken. The sample size listed for the ELISA experiments represent biological replicates.

## Acknowledgements

hiPSCs containing the PS1 ΔE9 mutation and the PS1M146V, APP$^{swe}$ mutations were the kind gifts of L Goldstein (University of California, San Diego, School of Medicine) and M Tessier-Lavigne (Rockefeller University and Stanford University), respectively. This work was supported in part by NIH grants P01 HD29587, DP1 DA041722, R01 NS086890, R01 AG056259, RF1 AG057409, and NINDS Core grant P30 NS076411 to SAL.

## Additional information

### Funding

| Funder | Grant reference number | Author |
| --- | --- | --- |
| National Institutes of Health | P01 HD29587 | Stuart A Lipton |
| National Institute of Neurological Disorders and Stroke | Core grant P30 NS076411 | Stuart A Lipton |
| National Institutes of Health | DP1 DA041722 | Stuart A Lipton |
| National Institutes of Health | R01 NS086890 | Stuart A Lipton |
| National Institutes of Health | R01 AG056259 | Stuart A Lipton |
| National Institutes of Health | RF1 AG057409 | Stuart A Lipton |

The funders had no role in study design, data collection and interpretation, or the decision to submit the work for publication.

### Author contributions

Swagata Ghatak, Conceptualization, Data curation, Formal analysis, Validation, Investigation, Visualization, Methodology; Nima Dolatabadi, Data curation, Methodology; Dorit Trudler, Methodology; XiaoTong Zhang, Yin Wu, Madhav Mohata, Data curation; Rajesh Ambasudhan, Conceptualization, Supervision; Maria Talantova, Conceptualization, Data curation, Formal analysis; Stuart A Lipton, Conceptualization, Resources, Supervision, Funding acquisition

### Author ORCIDs

Dorit Trudler (ID) http://orcid.org/0000-0002-5835-3322
Stuart A Lipton (ID) https://orcid.org/0000-0002-3490-1259

Decision letter and Author response
Decision letter https://doi.org/10.7554/eLife.50333.sa1
Author response https://doi.org/10.7554/eLife.50333.sa2

## Additional files

### Supplementary files
• Transparent reporting form

### Data availability
All data generated or analyzed during this study are included in the manuscript and supporting files.

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
