## [Decision Letter]

**Acceptance summary:**

In this manuscript, the authors show evidence that iPSC-derived neurons harboring APP or PS1 familial AD mutations display hyperexcitability both in 2D and 3D organoid culture conditions. The authors further explored the underlying mechanisms, some or all of which may contribute to hyperexcitability in AD neurons. These include decreased neurite length, increased sodium current density, and excitatory and inhibitory network imbalances at both synaptic and neuronal levels. This study represents an important step toward characterizing the underlying cell biological changes that may contribute to the pathophysiology of brain dysfunction in familial AD.

**Decision letter after peer review:**

Thank you for submitting your article "Mechanisms of hyperexcitability in Alzheimer's disease hiPSC-derived neurons and cerebral organoids vs. isogenic control" for consideration by *eLife*. Your article has been reviewed by three peer reviewers, and the evaluation has been overseen by a Reviewing Editor and Huda Zoghbi as the Senior Editor. The following individuals involved in review of your submission have agreed to reveal their identity: Lisa M Ellerby (Reviewer #1).

The reviewers have discussed the reviews with one another and the Reviewing Editor has drafted this decision to help you prepare a revised submission.

Summary:

In this manuscript, the authors generate 2D and 3D organoid cultures of IPSC-derived neurons harboring APP and PS1 familial AD mutations for electrophysiological studies. Compared with isogenic control neuron, the AD neurons showed hyperexcitability, and the authors showed evidence for cellular mechanisms underlying these excitability changes including decreased neurite length, increased sodium current density current density and excitatory and inhibitory network imbalances at synaptic and neuronal levels. Major strengths of the manuscript include the use of multiple human AD iPSC neurons with different familial AD mutation as well as previously validated isogenic control neurons, the use of organoid models to confirm the hyperexcitability of AD neurons in a brain-like environment, and the careful electrophysiological analyses. However, all three reviewers felt the study required mechanistic testing of the link between the hyperexcitability and dysregulation of APP processing by determining the effects of treating the cultures with BACE and/or γ-secretase inhibitors.

Essential revisions:

1) The authors need to discuss caveats of using iPSC-derived, developing neurons as a model to study a degenerative disorder of aging brains.

It is becoming increasingly clear in the literature that iPSC-derived neuron cultures and cerebral organoids represent very early stages of fetal brain development. Hence the degree that phenotypes reported in these neurons represent mechanisms relevant to the pathophysiology of diseases of brain aging are unclear. Specifically, whether the hyperactivity measured in the cultures in this study are caused by similar pathophysiological mechanisms that give rise to hyperexcitability in aged AD brains is unknown and, at the current time, unknowable. Despite this caveat, the reviewers all agree that iPSC-derived neuron studies like the one in this manuscript are still significant and will be of broad interest to the AD field because this is one major new approach to discovering cellular consequences of AD-associated genetic mutations. However, the authors must discuss the impact of using this reductionist experimental system on the strength of their conclusions for AD in much greater detail in the manuscript. Specifically, they should more directly address the alternative that there are developmental explanations for the hyperexcitability data shown here that could be different from the causes of hyperexcitability in the degenerating AD brain. As one reviewer stated, "Although it is interesting that familial AD mutations in two different AD genes, APP and PS1, commonly induce the hyperexcitability in human embryonic neurons it is not clear if the embryonic hyperexcitability will trigger AD pathogenies after 40-60 years. There is no report that FAD patients harboring these mutations develop seizures or other hyperexcitability symptoms at the very early ages (like 1-2 years).".

– This issue should be directly addressed in the Discussion section to help the readers understand the caveats as well as the power of the iPSC approach.

– The authors should also be careful with their statement of conclusions (abstracting from embryonic neurons to aged brains) throughout the manuscript with this concern in mind.

2) Further validation data must be presented in this manuscript to demonstrate that the cells used in the study are what they are described to be.

– The authors must provide validation of mutations and their correction in the isogenic control lines even though they are using previously published lines. It is imperative to know that the authors validated in their own hands the key genetic mutations in the cells they were actually using.

– Further, one reviewer questioned why AD neurons with APP Swedish mutation showed a dramatic increase in Aβ_42_/Aβ_40_ ratio. The APP Swedish mutation increases BACE1-mediated APP cleavage and therefore increases C99, total Aβ_40_ and Aβ_42_ levels, not the Aβ_42_/Aβ_40_ ratio. The reviewer mentioned this result was at odds with a recent paper from the Tessier-Lavigne lab (Kwart et al., 2019; Figure 4B), which according to the methods of this paper does appear to be using the exact same lines. The authors need to explain if this is indeed different and confirm the APP Swedish cells used in this study.

3) The authors need to determine whether AD-mutation associated changes in APP processing contribute to the mechanism of the cellular changes that underlie hyperexcitability.

The premise of the study is that the genetic mutations of APP or PS1 in these lines are the cause of the cellular phenotypes. By showing Aβ ratios in Figure 1—figure supplement 1 the authors appear to be implying that the phenotypes could arise as a result of impaired APP processing. However, they never explicitly test this (or any other) mechanism.

– Specifically the authors should treat their iPSC-derived mutant and isogenic control neurons with inhibitors of BACE or γ-secretase and determine the effects on core phenotypes reported here.

– The authors should also elaborate on their hypotheses of what mechanisms might link the genetic mutations to cellular hyperexcitability. Part of this framing could include discussion of other iPSC derived neuron studies for AD – for example the recent paper from the Tessier-Lavigne lab mentioned above using some of the same lines here (Kwart et al., 2019).

– One reviewer suggested that the word "mechanism" be removed from the title to better reflect the largely correlational data in the manuscript.

---

## [Author Response]

Essential revisions:1) The authors need to discuss caveats of using iPSC-derived, developing neurons as a model to study a degenerative disorder of aging brains.It is becoming increasingly clear in the literature that iPSC-derived neuron cultures and cerebral organoids represent very early stages of fetal brain development. Hence the degree that phenotypes reported in these neurons represent mechanisms relevant to the pathophysiology of diseases of brain aging are unclear. Specifically, whether the hyperactivity measured in the cultures in this study are caused by similar pathophysiological mechanisms that give rise to hyperexcitability in aged AD brains is unknown and, at the current time, unknowable. Despite this caveat, the reviewers all agree that iPSC-derived neuron studies like the one in this manuscript are still significant and will be of broad interest to the AD field because this is one major new approach to discovering cellular consequences of AD-associated genetic mutations. However, the authors must discuss the impact of using this reductionist experimental system on the strength of their conclusions for AD in much greater detail in the manuscript. Specifically, they should more directly address the alternative that there are developmental explanations for the hyperexcitability data shown here that could be different from the causes of hyperexcitability in the degenerating AD brain. As one reviewer stated, "Although it is interesting that familial AD mutations in two different AD genes, APP and PS1, commonly induce the hyperexcitability in human embryonic neurons it is not clear if the embryonic hyperexcitability will trigger AD pathogenies after 40-60 years. There is no report that FAD patients harboring these mutations develop seizures or other hyperexcitability symptoms at the very early ages (like 1-2 years).".– This issue should be directly addressed in the Discussion section to help the readers understand the caveats as well as the power of the iPSC approach.– The authors should also be careful with their statement of conclusions (abstracting from embryonic neurons to aged brains) throughout the manuscript with this concern in mind.

We thank the reviewers for their suggestion. We have now added a paragraph in the Discussion section to address the caveats of using iPSC-derived neurons to study degenerative disorders of aging brains and have also highlighted how it can be useful for studying certain aspects of the disease (Discussion paragraph two). For example, the fact that the hyperexcitability of the AD hiPSC-derived neurons is similar to that seen in adult human AD brains, as cited in the manuscript does suggest that this feature of the pathology can be studied under our conditions.

2) Further validation data must be presented in this manuscript to demonstrate that the cells used in the study are what they are described to be.– The authors must provide validation of mutations and their correction in the isogenic control lines even though they are using previously published lines. It is imperative to know that the authors validated in their own hands the key genetic mutations in the cells they were actually using.

As the reviewers suggested, we have validated the genetic mutations in the neural progenitor cells we are using by genome sequencing. We show these results in Figure 1—figure supplement 1A.

– Further, one reviewer questioned why AD neurons with APP Swedish mutation showed a dramatic increase in Aβ_42_/Aβ_40_ ratio. The APP Swedish mutation increases BACE1-mediated APP cleavage and therefore increases C99, total Aβ_40_ and Aβ_42_ levels, not the Aβ_42_/Aβ_40_ ratio. The reviewer mentioned this result was at odds with a recent paper from the Tessier-Lavigne lab (Kwart et al., 2019; Figure 4B), which according to the methods of this paper does appear to be using the exact same lines. The authors need to explain if this is indeed different and confirm the APP Swedish cells used in this study.

We thank the reviewers for pointing this out. We now have used a similar ELISA kit as that employed by the authors of Kwart et al., 2019. In agreement with those authors, we find that total Aβ (Aβ_40_, Aβ_42_ and Aβ_38_) is increased in the APP^swe^/WT lines, while the ratio does not change significantly. We show these new results in Figure 1—figure supplement 1D, E.

The results we had reported in the original version of the manuscript may have been different because we used two different kits to analyze Aβ_40_ and Aβ_42_. The kit for detecting Aβ_42_ levels was an ultrasensitive ELISA kit while that for detecting Aβ_40_ levels was an ELISA kit of normal sensitivity. This could have resulted in underestimation of the differences in Aβ_40_ levels in the various samples. In the Meso Scale Discovery kit used in the revised version of the manuscript, the samples were loaded in a well that has four separate spots for detecting Aβ_40_, Aβ_42_ and Aβ_38_ levels. This decreases handling errors and also kit-to-kit variability. Additionally, we now normalize Aβ levels to the total protein content of the lysate, and have used this approach to estimate total Aβ levels in the various samples.

3) The authors need to determine whether AD-mutation associated changes in APP processing contribute to the mechanism of the cellular changes that underlie hyperexcitability.

*The premise of the study is that the genetic mutations of APP or PS1 in these lines are the cause of the cellular phenotypes. By showing A*β *ratios in Figure 1—figure supplement 1 the authors appear to be implying that the phenotypes could arise as a result of impaired APP processing. However, they never explicitly test this (or any other) mechanism.*

– Specifically the authors should treat their iPSC-derived mutant and isogenic control neurons with inhibitors of BACE or γ-secretase and determine the effects on core phenotypes reported here.

We thank the reviewers for suggesting the experiments with BACE and γ-secretase inhibitors. Using this approach, we found that the γ-secretase inhibitor (CE) decreases hyperexcitability in PS1 mutant (M146V/WT) hiPSC-derived neurons. Similarly, we found that a BACE1 inhibitor decreases hyperexcitability in APP Swedish mutant (APP^swe^/WT) hiPSC-derived neurons. We observed decreases in spontaneous action potentials, sodium current density, and EPSC frequency. We have added these results to the main manuscript as a new figure (Results subsection “γ-Secretase or BACE1 inhibition reverses the hyperexcitability of AD neurons” and Figure 6). Our results suggest that APP processing plays a role in the hyperexcitability observed in hiPSC-derived neurons bearing the FAD mutations used in this study. Again, we thank the reviewers for this suggestion.

– The authors should also elaborate on their hypotheses of what mechanisms might link the genetic mutations to cellular hyperexcitability. Part of this framing could include discussion of other iPSC derived neuron studies for AD – for example the recent paper from the Tessier-Lavigne lab mentioned above using some of the same lines here (Kwart et al., 2019).

We have added a paragraph about the various mechanisms that could link the genetic mutations to cellular hyperexcitability in the Discussion section as suggested by the reviewers (paragraph four subsection “Synaptic dysfunction contributes to aberrant activity in AD neurons”).

– One reviewer suggested that the word "mechanism" be removed from the title to better reflect the largely correlational data in the manuscript.

With our new data from the aforementioned experiments suggested by the reviewers, we now provide mechanistic insight into the physiology underlying the observed hyperexcitability. Additionally, we show that shorter neurites, increased presynaptic vesicular glutamate, decreased presynaptic vesicular GABA, and increased sodium current density all contribute to the hyperexcitable phenotype of the Alzheimer’s lines compared to their isogenic controls.

In the revised version, we also show that inhibitors of enzymes that process APP can rescue the hyperexcitability phenotype for all the Alzheimer’s lines, thus providing a link between the actual genetic mutation and the altered physiology of these neurons. Hence, we think the revised manuscript and new data now provide mechanistic insight into the abnormal pathophysiology observed in Alzheimer’s disease; accordingly, we suggest retaining the word “mechanism” in the title.